# Research on Structural–Mechanical Properties during the Castor Episperm Breaking Process

**Liu Yang** [1,*] , **Huan Chen** [1] , **Junyu Xiao** [1] , **Yuchao Fan** [1] , **Shaoyun Song** [1,2] , **Yonglin Zhang** [1,2] and **Xiaopeng Liu** [3]

1    College of Mechanical Engineering, Wuhan Polytechnical University, Wuhan 430048, China; ch2293305261@163.com (H.C.); xJY1289956642@163.com (J.X.); f805884856@163.com (Y.F.); wuli9228@163.com (S.S.); yin2482476406@163.com (Y.Z.)
2    Hubei Cereals and Oils Machinery Engineering Center, Wuhan 430048, China
3    School of Animal Science and Nutritional Engineering, Wuhan Polytechnic University, Wuhan 430048, China; gtyl0622@163.com
*    Correspondence: yangliuvictry@whpu.edu.cn

**Abstract:** Products from castor seeds have been widely used in various fields. In order to study the breaking behavior and rupture mechanism of castor seed episperm during coat shelling process, the force-structure property of coating castor seed was investigated by a self-developed texture analyzer with in situ optical microscopic observation. Influences of compression distance, velocity and working temperature were studied. The results showed that castor seed episperm rupture commonly happened from the tail end to the first end. Compression distance effect can change the episperm cracking degree. Under pressing distance 2–3 mm, the episperm easily cracked into two flaps, and the breaking force stabilized at 77 N. Pressing velocity has no significant effect on episperm breaking. Temperature changes the physical property. With an increase in temperature, breaking force presents a "slope" decline; under a temperature of 120 °C, temperature effect on the breaking force decreased significantly and the breaking force fell to about 52 N. The research results can provide theoretical basis for the castor episperm peeling.

**Keywords:** castor episperm; mechanical property; breaking force; curve fitting

## 1. Introduction

Castor is a kind of Euphorbiaceae plant. Castor seed oil content after shelling is extremely high, about 50% [1]. Castor oil is mainly stored in castor beans, wrapped in the seed shell. Castor bean is an important economic crop and a special industrial oil source crop [2]. It plays an extremely important role as a raw material for lubricating oil. In addition, castor bean has been widely used in the fields of bio-energy, medicine, cosmetics industry, etc. [3–5]. To better design the huller equipment for outer seed coat with minimum energy performance and less damage, it is important to fully understand the mechanical–structural behavior of castor seed coat under compression loading [6].

Castor oil is generally produced by a combination of mechanical extraction and solvent extraction [7]. During the process of spiral pressing, the castor episperm is destroyed first, and the damage to the castor seed kernel is closely related to the damage of the shell [8]. Therefore, the study of castor seed mechanical properties is helpful for the development and improvement of castor seed processing machinery design [9]. Until now, few studies on the mechanical properties of castor seed episperm were done; more extensive studies have been done on the castor seed's pressing property without a coat. At the same time, it provides theoretical guidance for oil plants' mechanical property research.

Lorestani tested the physical properties of castor seed, including Young's modulus, maximum force, and required energy for initial fracture at the yield point for castor seed based on geometric size [10,11]. Bircan investigated loading speed effect on castor seed's mechanical properties; the rupture force, energy, apparent elastic modulus, and toughness

were tested with a material testing machine. These properties are used to design and improve related machines such as expellers that are used for the extraction of oil from castor seeds [12]. Herak [13] studied the mechanical properties of leprosy seeds at different maturity stages and temperatures. Braga examined the basis of shell rupture needed for the development of new methodologies or techniques to reduce the drying period and to obtain a more efficient kernel extraction process. A damage evaluation of compressed nuts showed a trend for the void between kernel and shell to be the greatest at the position where the force acts longitudinally through the hilum [14]. Cristiana researched the hazelnut's physical and mechanical properties, and optimized specimen shapes to obtain stress to failure and elastic modulus values for both the kernel and shell [15]. Aviara tested some nut strength properties at different moisture contents under lateral and longitudinal loading orientations, using a Universal Testing Machine (UTM). Results showed that all the nut strength properties (bio-yield point, yield point, rupture point, bio-yield strength, rupture strength, compressive strength, modulus of elasticity, modulus of resilience, and modulus of toughness) decreased significantly with an increase in moisture content. Loading along the longitudinal axis had higher strength property values than lateral loading [16].

In recent years, research on agricultural products mechanical properties is extensive. Research on castor seed's mechanical properties is seldom done, and the mechanical–structural research on castor seed breaking and shelling is important. In this paper, castor seed with coat breaking behavior was researched with a self-developed texture analyzer with online observation. The effects of pressing distance, velocity, and temperature were deeply investigated. The castor seed shell breaking process and structural damage were analyzed in order to provide guidance for castor bean shelling and oil pressing.

## 2. Materials and Methods

### 2.1. Experimental Material

Castor seed with full granule and no damage was screened with an optical microscope. The same size object was selected as the research object: 14.0–15.5 mm in length and 8.7–9.2 in thickness. It was produced in Anhui Province in early 2021 and stored in a cool and dry place.

### 2.2. Experimental Device and Facility

A developed test rig was used to investigate the castor episperm mechanical–structural property. The working schematic diagram is shown in Figure 1. The test system mainly contains four components: the vertical displacement measurement part, the vertical loading part, the force measurement part, and the in situ observation part. A computer is the central part for controlling, testing, and data collecting. The principle and operation steps are detailed as follows. A castor seed is placed on the base. The seed stays stationary during the test. The seed compression force is recorded when seed contact interaction reaches the inductive force. An electric heating furnace is applied to get different working temperature environments.

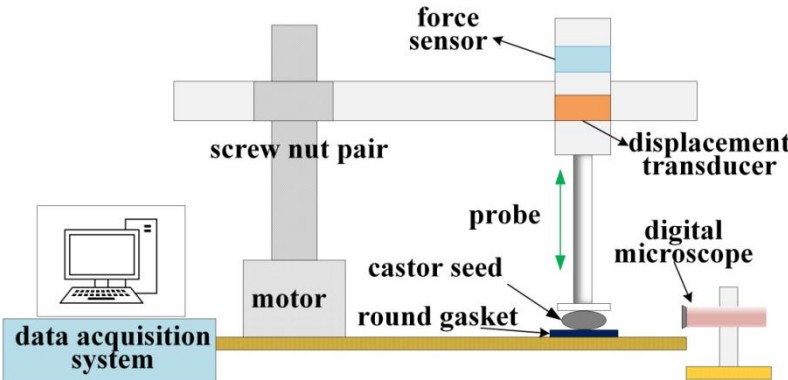

**Figure 1.** Schematic diagram of principle of in situ observation composite texture instrument.

In this experiment, the effects of pressing distance, velocity, and temperature on the cracking behavior of castor episperm were studied. Under the same condition, the test was completed on a developed texture testing instrument. The experimental samples were strictly brushed castor seeds with a geometric deviation of between 5% and 10%. Five repeated tests under each condition were conducted to ensure the data reliability. Table 1 showed the specific experimental parameter. When preparing samples under temperature conditions, castor seed samples were placed in a sealed glass container to exclude other influencing factors, heated in an electric temperature-controlled heating furnace, and then kept for 30 min at the objective temperature.

**Table 1.** Factors and levels of compression test.

| Variables (Level) | Pressing Distance (mm) | Pressing Velocity (mm/s) | Temperature (°C) |
|---|---|---|---|
| 1 | 1.0 | 0.05 | 20 |
| 2 | 1.5 | 0.10 | 60 |
| 3 | 2.0 | 0.15 | 90 |
| 4 | 2.5 | 0.20 | 120 |
| 5 | 3.0 | 0.25 | 150 |
| 6 | 4.0 | —— | —— |
| 7 | 5.0 | —— | —— |

A cylindrical probe with diameter 12.65 mm was selected for the breaking test. To ensure a more comprehensive compression, a glass disc with diameter of 31.70 mm was bonded on the probe contact surface with super adhesive glue. Before each test started, we adjusted the probe position and the bearing platform so that it was centrally aligned, and calibrated the height of probe and sensor settings. The probe inductive force was set at 0.05 N, probe moving velocity before and after the test was set at 1.0 mm/s and between 0.05–0.25 mm/s during the test, the in-pressing distance was set at 1–5 mm. The experimental method was set as a single test. The cracking of the episperm during compression was observed and recorded by a super-eye microscope, and the force–time curve was finally obtained in the data collection center. The texture apparatus schematic diagram is shown in Figure 1.

## 3. Results and Discussion

### 3.1. Extrusion Pressure-Time Curve and In-Situ Observation of Castor Seed Coat

Mechanical properties are the properties of a metal material to resist damage under load. For example, investigating the alloys mechanical properties, the stress–strain relationship is generally discussed due to the stability of their compressive contact area [17–19]. Due to the irregularity of plant seed morphology, the mechanical properties are generally characterized by the initial shell breaking force, displacement, or compression force. Referring to the curve treatment in the literature [20], the castor seed was compressed under the condition of in-pressing distance 4 mm, pressing velocity 0.1 mm/s, and room temperature 20 °C, and the force–time curve was obtained as shown in Figure 2. The pressing curve initial stage showed that the probe just contacted the castor episperm when the inductive force reached 0.05 N. With the probe continuously pressing downward, the episperm was subjected to increasing pressure. At the beginning, when the episperm was deformed by extrusion, the gap between outer seed coat and inner seed coat decreased. When the compression force increased to the maximum point 'a' 82.7 N, it suddenly dropped to point 'b'; the pressing force at this point was 18.4 N. The extrusion force started to fluctuate and slowly rose to point 'c'. The seed extrusion force sharp decreasing from 'a' to 'b' was mainly due to the crack happening for seed episperm plastic deformation accumulation. In addition, the outer seed coat and inner seed coat are not completely attached, a certain gap exists between them, thus causing the extrusion force to suddenly drop during continuous compression. With the compression force continuously increasing, the castor episperm

began to break and compression force also decreased. At this time, the load on the episperm is called "rupture load", namely the shell breaking force (Herak, 2010). The increasing compression force from point 'b' to point 'c' is the result of the castor seed outer seed coat and endosperm extruding process. The force fluctuation in zone 'd' is due to crack on the episperm and damage to the endosperm. It can be seen from the curve that the breaking force of castor outer seed coat is 82.7 N.

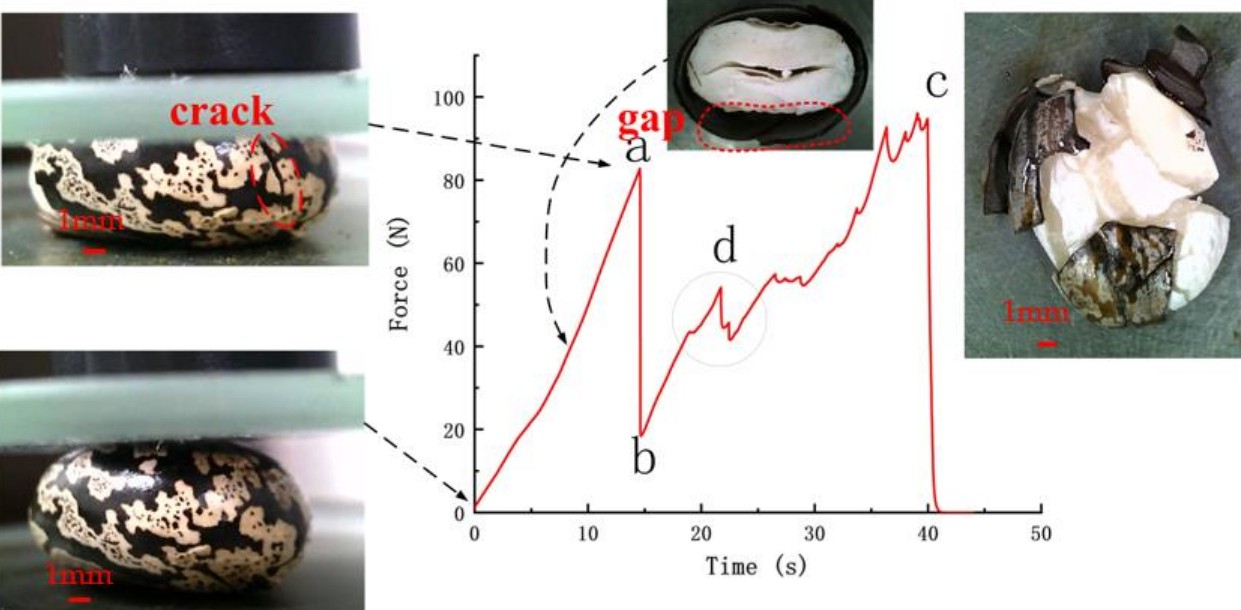

**Figure 2.** Force–time curve during extrusion.

### 3.2. Mechanical Properties of Castor Seed Coat under Different Pressure Spacing

Under different compression distances (1.0, 1.5, 2.0, 2.5, 3.0, 4.0, and 5.0 mm), the episperm compression test was carried out at 0.1 mm/s and laboratory temperature 20 ± 2 °C. Data was collected when the compression force suddenly decreased (point 'a' in Figure 2). The maximum force happening point before it decreased was selected to obtain compression force–pressing distance curve data. Figure 3 showed that when pressing distance increased from 1 mm to 1.5 mm, the compression force on the episperm increased rapidly from 58 N to 83 N. When the pressing distance increased from 1.5 mm to 4 mm, the compression force gradually decreased to 73 N with a low rate in the middle segment, and the compression force remained stable at about 77 N. When the pressing distance increased to 5 mm, the compression force began to rise sharply to 91 N.

The outer seed coat deformation diagram under different pressing distances was obtained by optical observation microscope, as shown in Figure 3. It was found that when the probe compression distance reached 1 mm, the outer seed coat showed obviously elastic deformation and no significant change happened on the seed coat. The compression force increased with pressing distance increasing. Under a pressing distance of 1.5 mm, a few cracks appeared on the seed coat tail area and the compression force reached the shell breaking point. With the probe continuously pressing downward, the castor episperm was close to the inner seed coat and the coat breaking force reduced for the coat inside and outside contact effect. Under pressing distances between 1.5–2 mm, the rupture area was only partially cracked; the cracking force decreased significantly during this period. With pressing distances between 2–3 mm, the episperm was basically broken, as shown in Figure 4. The contact part between episperm and the inner seed coat were closely adhered to each other in a large area; the reduction of shell breaking force was deceased. When pressing distance increased to 5 mm, the episperm was compressed rapidly in a short time,

and this produced all-around rupture, as shown in Figure 4. The episperm and inside seed were also compressed into a cake shape. The shell breaking force was relatively large.

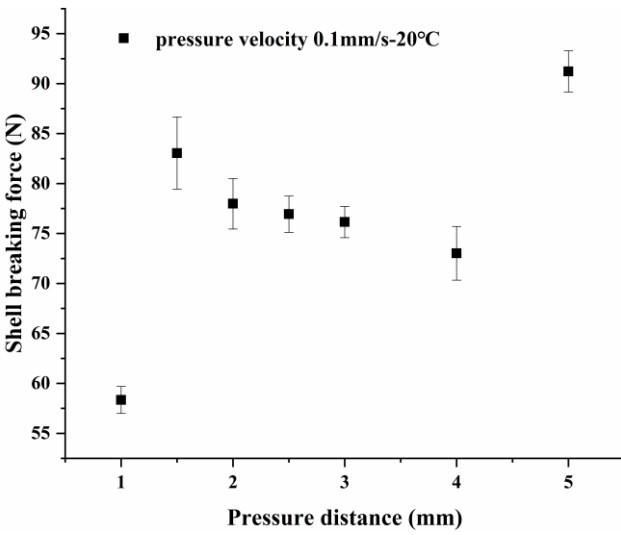

**Figure 3.** Pressing distance–mechanical properties of shell breaking force.

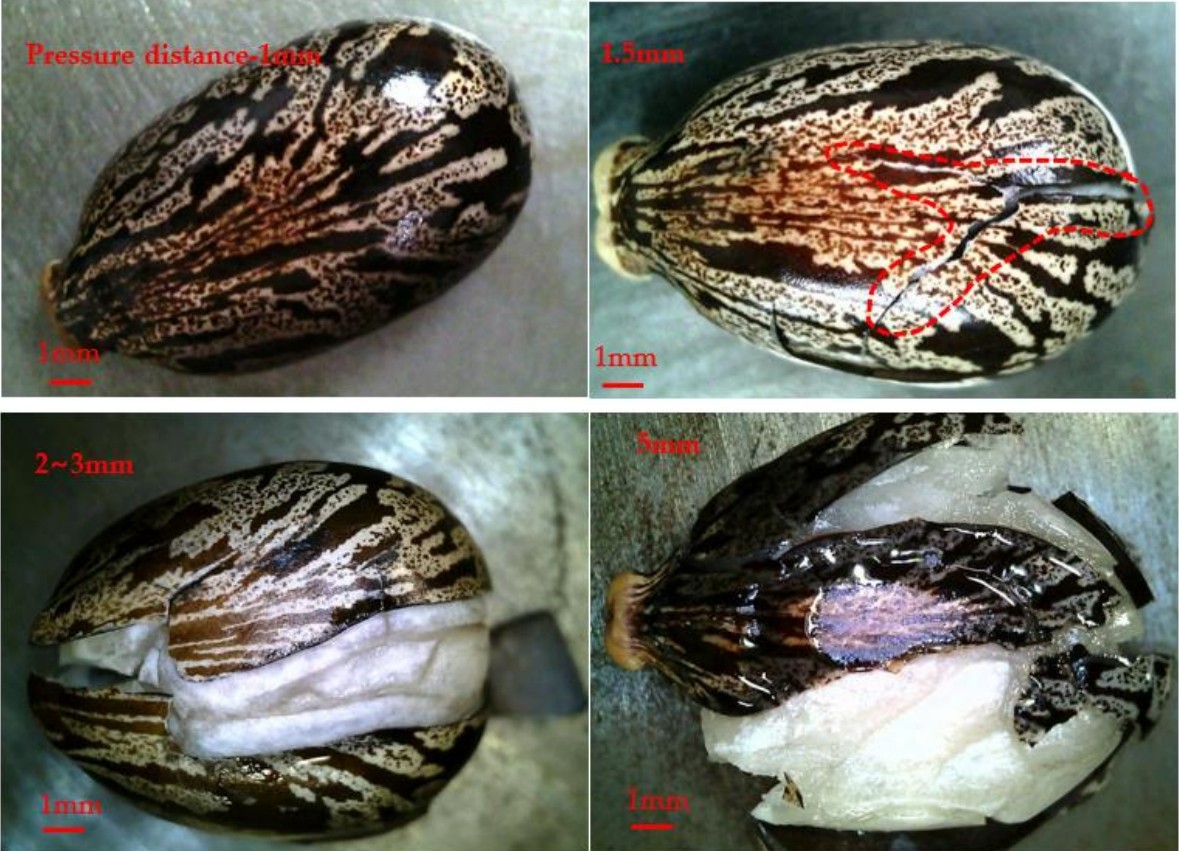

**Figure 4.** Observation image of cracks in castor bean outer seed coat.

### 3.3. Mechanical Property of Castor Outer Seed Coat under Different Pressing Rates

Under different pressing velocities (0.05, 0.10, 0.15, 0.20, and 0.25 mm/s), the compression force on episperm was tested under distance 3 mm and 20 ± 2 °C. The compression force abrupt-change point was collected to obtain the force–velocity curve, which is shown in Figure 5. It can be seen that the shell breaking force on the castor episperm fluctu-

ated within a certain range without obvious influence, and the average value of the shell breaking force was about 72 N.

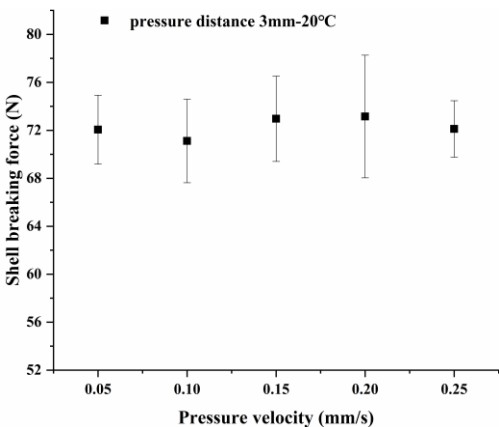

**Figure 5.** Pressing velocity–shell breaking force mechanical characteristics diagram.

*3.4. Mechanical Properties of Castor Bean Outer Seed Coat at Different Temperatures*

Temperature can change the physical properties of castor bean; it is one of the important factors affecting the mechanical property of castor episperm. Figure 6 showed the castor shell breaking force curve under temperatures (20 °C, 60 °C, 90 °C, 120 °C, and 150 °C) with compression distances 3 mm and 0.1 mm/s. In addition, cross-over tests at different temperatures at pressure distances (1.5 mm, 3.0 mm, and 5.0 mm) and pressure velocities (0.05 mm/s, 0.10 mm/s, and 0.20 mm/s) were completed in order to actually conduct a comprehensive evaluation of these factors. According to the pressing distance 3 mm-velocity 0.10 mm/s curve, the castor seed shell breaking force showed a "slope" decline with the increase in temperature. When the temperature was below 60 °C, the change of shell breaking force was not obvious and the shell breaking force was about 71 N. When the temperature was higher than 60 °C, the breaking force began to decrease rapidly and was close to stable. At this time, increasing the temperature above 60 °C had no significant effect on the breaking force. The castor seed shell breaking force under 150 °C was the smallest, about 52 N. The remaining curves were observed and we found that curve 7, under pressing distances 5.0 mm and 0.20 mm/s, had the highest breaking force at 20 °C with 77.3 N. Curve 3, with pressing distance 1.5 mm and 0.05 mm/s, had the lowest breaking force with 70 N. For temperatures between 20 °C and 120 °C, the episperm breaking force showed a trend of maintaining a stable first after a sharp decline. A change in the shell breaking force increased after temperatures exceeding 120 °C, and the breaking force growth trend of curves 3–8 is obviously different from that of curves 1 and 2.

The reason is that temperatures around 60 °C had no obvious effect on the episperm's physical property, and the shell breaking force at this temperature was almost the same as at room temperature. Yet, the combined effect of pressure distance and pressure velocity leads to significant fluctuations in the breaking force between groups of curves at the same temperature, with curve 7 having the highest breaking force. When the temperature exceeds 60 °C, the higher the temperature, the greater the influence on episperm, making episperm have a texture brittle that is easier to be destroyed; thus, the episperm breaking force gradually decreased. When the working temperature gets to be 120 °C, compared with the episperm observation image at 90 °C, it was obvious that the castor seed endosperm turned yellow partially, the damage or cracks decreased on the endosperm, the castor seed became hard and brittle at 120 °C, and the episperm carrying capacity naturally decreased, so the shell-breaking force decreased significantly. When the temperature exceeded 120 °C, the influence of temperature on the episperm reached its limitation, and the influence of temperature on its physical property and shell breaking force decreased significantly. However, taking into account Sections 3.2 and 3.3, it is understood that the pressing velocity has less effect on the episperm shell breaking force, and the pressure distance has a

greater effect on the shell breaking force, and the force relationship under different pressure distances: 5.0 mm > 1.5 mm > 3.0 mm. Therefore, the breaking force of curve 3–8 appeared to increase at 150 °C. In addition, it is also possible that when the temperature is so high the episperm material hardens while its tenacity is also improved, making it more resistant to fracture.

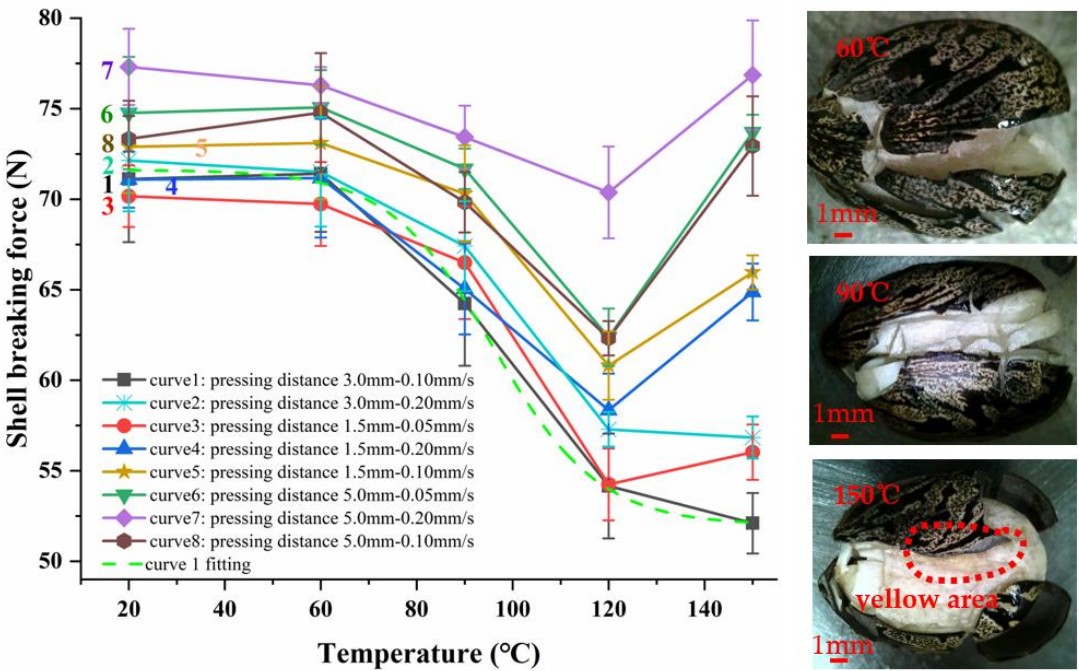

**Figure 6.** Temperature-shell breaking force curve and corresponding observation image for curve 1 under different temperatures.

Analyzing the combined effects of pressure distance and pressure velocity under different temperatures, it is found that the shell breaking force increased under the pressure distance too high or too low, and it is the smallest at a pressure distance of 3 mm. Curve 1 was the best test condition during the episperm shell breaking test. With regression analysis of the curve 1, the Boltzmann function was used to fit the relationship between the shell-breaking force and temperature on the episperm. The fitting expression was as follows:

$$y = b + (a - b)/(1 + \exp((x - x_0)/dx)), \tag{1}$$

where: $x$ is temperature and $y$ is the shell breaking force on a castor episperm. a = 71.40204; b = 51.96460; $x_0$ = 96.30092; and $R^2$ = 0.99924.

## 4. Conclusions

The castor seeds' mechanical and structural properties with seed coat was studied with a developed texture apparatus. The test results showed that pressing distance had a significant effect on the cracking degree and shell breaking force of castor outer seed coat. With regression analysis of the seed coat compression test, the Boltzmann function was used to obtain the relationship between the breaking strength and temperature. The castor seed test has a significant impact on the degree of cracking and breaking force. The larger the pressing distance, the more intense the crack propagation of episperm and the cracking force gradually decreased. When the episperm broke completely, the breaking force was stable at about 77 N. The effect of pressing velocity on the shell breaking force of the episperm is small. When the pressing velocity is larger than 0.25 mm/s, the shell breaking force of episperm will decrease. The shell breaking force of the episperm decreased "downhill" with an increase in temperature. When the temperature exceeded 120 °C, the attenuation of shell breaking force was stable at about 52 N.

**Author Contributions:** L.Y.: investigation, conceptualization, methodology, and funding. H.C.: investigation, writing—original draft. J.X. and Y.F.: analysis and materials. S.S.: review and editing. Y.Z.: data analysis, and review and editing. X.L.: experimental assistance. All authors have read and agreed to the published version of the manuscript.

**Funding:** The research is mainly sponsored by the Science Foundation of Wuhan Polytechnic University (No.2020J06, 2019RZ08) and Jiangsu Key Laboratory of Advanced Food Manufacturing Equipment & Technology (FM-202103); part of the research is supported by the Tribology Science Fund of State Key Laboratory of Tribology (No. SKLTKF14A08); and Youth Project of Natural Science Foundation of Hubei Province (No. 2020CFB436).

**Institutional Review Board Statement:** Not applicable.

**Informed Consent Statement:** Not applicable.

**Data Availability Statement:** The datasets generated and analyzed during the current study are available from the corresponding author on reasonable request.

**Acknowledgments:** Thanks very much for the support of Hubei Cereals and Oils Machinery Engineering Technology Research Center at Wuhan Polytechnic University.

**Conflicts of Interest:** The authors declare that they have no conflict of interest.

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
