# Peer review of "Research on Structural–Mechanical Properties during the Castor Episperm Breaking Process"

_processes, doi:10.3390/pr9101777_

Round 1
Reviewer 1 Report
The authors reported the analysis of the mechanical property when pressing castor seeds. Despite the effort make by the authors, I have some major concerns that make me reject this manuscript for publication.
- More importantly, the whole study lacks of scientific soundness and seems to be incompleted, given that the authors reported to have conducted a comprehensive analysis of the pressing parameters "Effect of pressing distance, speed and temperature was deeply investigated". However, after reading the manuscript I regret to say that is not the case. Moreover, below you can find some other comments that I hope may help you improve the quality of the study for further applications.
- Numbering of the titles should be revised (Lines 77 and 82).
- Lines 89-90. How did the authors conduct the preconditioning of the samples? That should be included (more detailed information).
- Lines 94-95. “The experimental samples were castor seeds of similar geometrical size, the shape and size of castor seeds were slightly different”. The authors contradict themselves. The sizes and size’s deviations should be included.
- Lines 146-147 are just some examples, but the text require a comprehensive English language revision.
- Table 1. Were there only 7 tests? Given that according to this manuscript the authors conducted several tests in order to analyse the influence of the pressure spacing at a constant speed, the influence of pressing rates at a constant pressure distance and so on, all tests should be clearly shown, otherwise from table 1 it could be missunderstood.
- Section 3.4 should also include the analysis of the temperature influence at different pressure distances and pressure speed in order to actually conduct a comprehensive evaluation of these factors.
Reviewer 2 Report
1. Some figures are missed such as Figs. 3, 5 and 6.
2. Mechanical property (transferred into stress such as MPa) should be compared with some other materials such as Al, Mg or polymer materials and the corresponding analysis should cite some references. These refetences could be referred such as: Rare Met., 2020. doi 10.1007/s12598-020-01474-6, Rare Met., 2019;38(1):42-51, Rare Met., 2020;39(10): 1127–1133, Rare Met., 2020;39(5):562-569, Journal of Plant Physiology 224–225 (2018) 56–67, Rare Met., 2019;38(6):561-570.
Round 2
Reviewer 1 Report
Although the authors have appropriately improved the quality of the manuscript. They have completed some of the serious flaws. However I still consider it lacks of a significant scientific soundness.